# Dimension Reduction for High-dimensional Small Counts with KL Divergence

**Yurong Ling**[1]  **Jing-Hao Xue**[1]

[1]Department of Statistical Science, University College London, London, UK

## Abstract

Dimension reduction for high-dimensional count data with a large proportion of zeros is an important task in various applications. As a large number of dimension reduction methods rely on the proximity measure, we develop a dissimilarity measure that is well-suited for small counts based on the Kullback-Leibler divergence. We compare the proposed measure with other widely used dissimilarity measures and show that the proposed one has superior discriminative ability when applied to high-dimensional count data having an excess of zeros. Extensive empirical results, on both simulated and publicly-available real-world datasets that contain many zeros, demonstrate that the proposed dissimilarity measure can improve a wide range of dimension reduction methods.

## 1 INTRODUCTION

High-dimensional count data, especially those with a large proportion of zeros, are omnipresent in various fields, such as ecology and genomics [Warton, 2018, Townes et al., 2019, Svensson, 2020]. Dimension reduction (DR) techniques are used to extract useful information from high-dimensional count data, by eliminating noisy/uninformative dimensions of the data. Owing to the mean-variance dependency that is often observed in count data, it is inappropriate to apply standard DR methods that are optimal under the normality assumption, such as principal component analysis (PCA) [Pearson, 1901, Hotelling, 1933, Tipping and Bishop, 1999] and Gaussian process latent variable model (GPLVM) [Lawrence, 2005], to the data.

Hence, to perform DR on count data, a number of specific strategies/methods have been proposed. A common strategy is to first apply a variance-stabilizing transformation (VST) to the data [Bartlett, 1947, Anscombe, 1948], aiming to make the data more Gaussian-like, and then feed the transformed data into standard DR approaches. The transformation function is specifically chosen to remove the mean-variance dependency. Popular transformation functions include the square root, logarithm, and inverse hyperbolic sine functions. Despite the widespread use of the VSTs, they can only be guaranteed to work well with large counts [Bartlett, 1947, Anscombe, 1948] and cannot reasonably be expected to stabilize the variance of small counts containing a large faction of zeros [Yu, 2009, Warton, 2018]. Rather than focusing on making count data more normally distributed, several approaches have been developed to directly model the original data. With the assumption that count data follow the exponential family distributions, PCA variants maximise the likelihood of the observed data to get the low-dimensional representation [Collins et al., 2002, Mohamed et al., 2008, Li and Tao, 2013, Smallman et al., 2020]. By adopting the same distributional assumption, a robust estimator of the covariance matrix is derived and the data of reduced dimension are obtained by the eigendecomposition of this estimator [Liu et al., 2018]. Nonnegative matrix factorization (NMF) acquires the low-dimensional representation by factorizing count data matrix into two nonnegative matrices of low rank [Dhillon and Sra, 2005, Févotte and Idier, 2011, Cichocki et al., 2011]. Despite the popularity of NMF and PCA variants, it is unclear whether they perform well on count data having an excess of zeros.

Unlike the aforementioned works, we focus on developing measures that can reliably quantify the pairwise dissimilarity for small count data, motivated by the importance of proximity matrix in common DR frameworks. Specifically, many DR approaches seek to preserve properties of a proximity matrix of high-dimensional data when reducing the dimension of the data. Examples of such approaches include PCA with the Euclidean distance matrix and the Gram matrix [Mardia et al., 1979, Lawrence, 2005], GPLVM with the Gram matrix [Lawrence, 2005], multidimensional scaling (MDS) with an input dissimilarity matrix [Torgerson, 1952], and t-distributed stochastic neighbour embedding (tSNE)

*Accepted for the 38th Conference on Uncertainty in Artificial Intelligence* (UAI 2022).

with the matrix of the Gaussian kernels [Maaten and Hinton, 2008]. Therefore, a proximity measure that properly quantifies the dissimilarity between small-count data points could benefit a wide range of DR methods.

The two core contributions of this paper can be summarized as follows. First, we develop two dissimilarity measures for small count data based on the Kullback-Leibler (KL) divergence [Kullback and Leibler, 1951] and the assumption that the data follow either Poisson or negative binomial (NB) distributions. We take both Poisson and NB distributions into account, as it is common to model count data with these two types of distributions [Zeileis et al., 2008, Chan and Vasconcelos, 2009, Lindén and Mäntyniemi, 2011, Townes et al., 2019, Kim et al., 2020]. Furthermore, to reliably calculate the KL divergence, we propose to use empirical Bayes estimators to estimate the distributional parameters. Secondly, we propose an index to evaluate the discriminative abilities of different dissimilarity measures and show that the measure developed with the NB assumption has superior discriminative ability compared with other widely used dissimilarity measures for high-dimensional small counts, in terms of their statistical behaviours. Moreover, consistent with our statistical investigation, the experimental results, on both real and simulated count data, also demonstrate that the measure obtained with the NB assumption is superior to other measures when handling small counts.

The rest of this paper can be summarized as follows. First, we present standard transformations for count data in Section 2.1. We then derive two new dissimilarity measures with the KL divergence and the empirical Bayes estimators in Section 2.2. Secondly, we propose an index which evaluates the discriminative ability of a dissimilarity measure (Section 3.1) and compare different dissimilarity measures according to the proposed index. It is shown that, when applied to small counts, the Euclidean distance of the transformed data exhibits better discriminative ability than the original Euclidean distance, although the corresponding VST is unable to stabilize the variances (Section 3.2). More importantly, the measure obtained with the NB assumption is expected to perform the best when used for separating different distributions of small count data (Section 3.3). Lastly, we present the experimental results of representative DR methods with different measures on both real and simulated datasets (Section 4).

# 2 DISSIMILARITY MEASURES FOR COUNT DATA

In this section, we first present widely used VSTs for count data and then derive two new dissimilarity measures with the KL divergence and the empirical Bayes estimators for small counts.

## 2.1 VARIANCE-STABILIZING TRANSFORMATIONS (VSTS)

A VST is a data transformation that applies to data such that the variance of the transformed data is independent of their mean. Most VSTs for count data are developed by assuming data follow an either Poisson or NB distributions [Bartlett, 1947, Anscombe, 1948]. Let $y$ be the raw counts. The square root transformation

$$g_r(y) = \sqrt{y + \frac{3}{4}} \tag{1}$$

is a popular technique for stabilizing the variance of a Poisson random variable. For an NB random variable $y$ which counts the number of successes and has the PMF $\binom{y+r-1}{y}(1-p)^r p^y$, where $p$ is the probability of success and $r$ represents the number of failures, a prevalent transformation is

$$g_{\text{asin}}(y) = \text{arcsinh}\sqrt{\frac{y + \frac{3}{8}}{r - \frac{3}{4}}}, \tag{2}$$

where arcsinh is the inverse hyperbolic sine function. Since $g_{\text{asin}}(y)$ requires an approximate knowledge of $r$ and in some cases it cannot be estimated well enough, a simpler logarithm transformation with a pseudocount 1 is preferred in practice, which is given by

$$g_{\text{log}}(y) = \log(y + 1). \tag{3}$$

As mentioned before, these transformations fail to stabilize the variance of small counts. Thus, there is no guarantee that the Euclidean distances of the data transformed from raw counts by these VSTs perform well on small counts.

## 2.2 TWO NEW DISSIMILARITY MEASURES DEVELOPED WITH KL DIVERGENCE

The KL divergence is a statistical measure of how one probability distribution is different from a second, reference probability distribution [Kullback and Leibler, 1951]. For discrete probability distributions $P$ and $Q$ defined on the same probability space $\mathbf{Z}$, the KL divergence is defined as $D_{\text{KL}}(P \mid Q) = \sum_{z \in \mathbf{Z}} P(z) \log \frac{P(z)}{Q(z)}$. The KL divergence for continuous random variables can be defined similarly by replacing the sum with the integral. For a pair of univariate normal distributions, $P : \mathcal{N}(\mu_x, \sigma^2)$ and $Q : \mathcal{N}(\mu_y, \sigma^2)$, we have $D_{\text{KL}}\left[\mathcal{N}(\mu_x, \sigma^2) \mid \mathcal{N}(\mu_y, \sigma^2)\right] = \frac{(\mu_x - \mu_y)^2}{2\sigma^2}$. The squared Euclidean distance $D_E^2$ between two vectors $\mathbf{x} = [x_1, \ldots, x_p]^T, \mathbf{y} = [y_1, \ldots, y_p]^T \in \mathbb{R}^p$ is equivalent to the sum of the KL divergence between two univariate normal distributions across dimensions up to a constant $\frac{1}{2\sigma^2}$ and the mean values of the distributions are estimated by the maximum likelihood estimators (MLEs). This equivalence

is shown by the following equation:

$$\sum_{i=1}^{p} \hat{D}_{\mathrm{KL}} \left[ \mathcal{N}(\mu_{ix}, \sigma^2) \mid \mathcal{N}(\mu_{iy}, \sigma^2) \right] = \sum_{i=1}^{p} \frac{(\hat{\mu}_{ix} - \hat{\mu}_{iy})^2}{2\sigma^2}$$

$$= \sum_{i=1}^{p} \frac{(x_i - y_i)^2}{2\sigma^2},$$

where $x_i$ and $y_i$ are the MLEs of mean parameters of the normal distributions on the $i$-th dimension of **x** and **y**, respectively, when there is only one realisation observed for each distribution.

Stimulated by the equivalence between $D_E^2$ and the KL divergence, we propose to quantify the pairwise dissimilarity for count data with the KL divergence. To calculate the KL divergence, the distribution type and the corresponding parameter values are required to be specified. For the distribution type, we assume the observed data follow either Poisson or NB distributions, which are commonly used for modelling count data. Regarding the parameter estimation, a straightforward estimator is the MLE. However, the use of MLE incurs a numerical problem in practice. To clarify this problem, we first derive two dissimilarity measures with the MLEs for Poisson and NB distributions, respectively. Suppose $x_i$ and $y_i$ follow $\mathrm{Pois}(\lambda_{ix})$ and $\mathrm{Pois}(\lambda_{iy})$, respectively. The respective MLEs of $\lambda_{ix}$ and $\lambda_{iy}$ are $x_i$ and $y_i$. The KL divergence between **x** and **y** with these MLEs is thus

$$\sum_{i=1}^{p} \hat{D}_{KL} \left[ \mathrm{Pois}(\lambda_{ix}) \mid \mathrm{Pois}(\lambda_{iy}) \right] = \sum_{i=1}^{p} \left[ y_i - x_i + x_i \log \frac{x_i}{y_i} \right]. \quad (4)$$

Analogously, we suppose $x_i$ and $y_i$ follow $\mathrm{NB}(r, p_{ix})$ and $\mathrm{NB}(r, p_{iy})$, respectively, with known $r$. Note that there are multiple definitions of the NB distribution and we use the following ones: the PMF of $\mathrm{NB}(r, p_{ix})$ for $x_i$ is $\binom{x_i + r - 1}{x_i}(1 - p_{ix})^r p_{ix}^{x_i}$ and similarly for $\mathrm{NB}(r, p_{iy})$, where $p_{ix}$ and $p_{iy}$ are the probabilities of success, $r$ represents the number of failures, and $x_i$ and $y_i$ count the numbers of successes. The respective MLEs of $p_{ix}$ and $p_{iy}$ are $\frac{x_i}{x_i + r}$ and $\frac{y_i}{y_i + r}$. The KL divergence between **x** and **y** with the MLEs is given by

$$\sum_{i=1}^{p} \hat{D}_{KL} \left[ \mathrm{NB}(r, p_{ix}) \mid \mathrm{NB}(r, p_{iy}) \right]$$

$$= \sum_{i=1}^{p} r \log \frac{y_i + r}{x_i + r} + x_i \log \frac{x_i(y_i + r)}{y_i(x_i + r)}. \quad (5)$$

The dissimilarity measures presented in Equation (4) and Equation (5) both involve the logarithm terms, and thus zeros in count data would result in the numerical problem. Further, since the MLEs are close to the true values of parameters only if the number of observations is sufficiently

large, the MLE calculated from one observation respectively are unreliable, so are $\hat{D}_{KL} \left[ \mathrm{Pois}(\lambda_{ix}) \mid \mathrm{Pois}(\lambda_{iy}) \right]$ and $\hat{D}_{KL} \left[ \mathrm{NB}(r, p_{ix}) \mid \mathrm{NB}(r, p_{iy}) \right]$.

To address these issues, we propose to use the empirical Bayes estimators rather than the MLEs. The conjugate priors of Poisson and NB distributions are employed for estimating the parameters ($\lambda_{ix}$, $\lambda_{iy}$, $p_{ix}$, $p_{iy}$). In addition, the hyperparameters of these priors are learned from data themselves, sidestepping the difficulty of specifying proper priors to some degree. Concretely, we specify a Gamma prior distribution $G(m_i, 1)$, where the shape parameter $m_i$ is the mean value of the $i$-th dimension across all data points and the other parameter is the scale parameter, for the Poisson means ($\lambda_{ix}$, $\lambda_{iy}$). For the probability parameters of NB distributions ($p_{ix}$, $p_{iy}$), we specify a Beta prior distribution $B(m_i, r)$ for them. Note that the mean value can be thought of an additional observation. With the priors, we obtain the posterior distribution of $\lambda_{ix}$ is $G(m_i + x_i, \frac{1}{2})$ and that of $\lambda_{iy}$ is $G(m_i + y_i, \frac{1}{2})$. The posterior means, which are $\frac{m_i + x_i}{2}$ and $\frac{m_i + y_i}{2}$, respectively, are used as the estimated distributional parameters. Analogously, we obtain the posterior mean $\frac{m_i + x_i}{m_i + x_i + 2r}$ from the posterior distribution $B(m_i + x_i, 2r)$ and $\frac{m_i + y_i}{m_i + y_i + 2r}$ from $B(m_i + y_i, 2r)$, as the estimated distributional parameters for NB distributions. Now we obtain the KL divergence between **x** and **y** with the Bayes estimators (posterior means):

$$\hat{D}_{KL}^{Bayes} \left[ \mathrm{Pois}(\lambda_{ix}) \mid \mathrm{Pois}(\lambda_{iy}) \right]$$

$$= \frac{1}{2} \sum_{i=1}^{p} y_i - x_i + (x_i + m_i) \log \frac{x_i + m_i}{y_i + m_i},$$

$$\hat{D}_{KL}^{Bayes} \left[ \mathrm{NB}(r, p_{ix}) \mid \mathrm{NB}(r, p_{iy}) \right]$$

$$= \sum_{i=1}^{p} \left[ r \log \frac{y_i + m_i + 2r}{x_i + m_i + 2r} \right.$$

$$\left. + \frac{x_i + m_i}{2} \log \frac{(x_i + m_i)(y_i + m_i + 2r)}{(y_i + m_i)(x_i + m_i + 2r)} \right]. \quad (6)$$

The logarithm terms in Equation (6) are well defined for $m_i > 0$, which is easily satisfied in practice as only meaningless features in the form of all zeros have $m_i = 0$. Owing to the asymmetry of the KL divergence, we propose to use

$$D_P^2 = \hat{D}_{KL}^{Bayes} \left[ \mathrm{Pois}(\lambda_{ix}) \mid \mathrm{Pois}(\lambda_{iy}) \right]$$

$$+ \hat{D}_{KL}^{Bayes} \left[ \mathrm{Pois}(\lambda_{iy}) \mid \mathrm{Pois}(\lambda_{ix}) \right]$$

$$D_{NB}^2 = \hat{D}_{KL}^{Bayes} \left[ \mathrm{NB}(r, p_{ix}) \mid \mathrm{NB}(r, p_{iy}) \right]$$

$$+ \hat{D}_{KL}^{Bayes} \left[ \mathrm{NB}(r, p_{iy}) \mid \mathrm{NB}(r, p_{ix}) \right] \quad (7)$$

to measure the pairwise dissimilarity for count data. Table 1 lists the dissimilarity measures that we take into account in this paper. Note that for $D_P$ and $D_{NB}$ we ignore their multiplicative constant $\frac{1}{2}$ for conciseness.

Table 1: Dissimilarity measures and their equations.

| Measure | Equation |
|---------|----------|
| $D_E(\mathbf{x}, \mathbf{y})$ | $\left[\sum_{i=1}^p (x_i - y_i)^2\right]^{\frac{1}{2}}$ |
| $D_r(\mathbf{x}, \mathbf{y})$ | $\left[\sum_{i=1}^p (g_r(x_i) - g_r(y_i))^2\right]^{\frac{1}{2}}$ |
| $D_{\mathrm{asin}}(\mathbf{x}, \mathbf{y})$ | $\left[\sum_{i=1}^p (g_{\mathrm{asin}}(x_i) - g_{\mathrm{asin}}(y_i))^2\right]^{\frac{1}{2}}$ |
| $D_{\log}(\mathbf{x}, \mathbf{y})$ | $\left[\sum_{i=1}^p (g_{\log}(x_i) - g_{\log}(y_i))^2\right]^{\frac{1}{2}}$ |
| $D_P(\mathbf{x}, \mathbf{y})$ | $\left[\sum_{i=1}^p (\log(x_i + m_i) - \log(y_i + m_i))(x_i - y_i)\right]^{\frac{1}{2}}$ |
| $D_{NB}(\mathbf{x}, \mathbf{y})$ | $\left[\sum_{i=1}^p \left(\log \frac{x_i + m_i}{x_i + m_i + 2r} - \log \frac{y_i + m_i}{y_i + m_i + 2r}\right)(x_i - y_i)\right]^{\frac{1}{2}}$ |

# 3 COMPARISON OF DISSIMILARITY MEASURES FOR HIGH-DIMENSIONAL SMALL COUNTS

In this section, we compare different measures listed in Table 1, according to their abilities to distinguish distributions that tend to produce small counts. First, we propose an index to quantify the discriminative abilities of different dissimilarity measures. Then, based on the proposed index, we investigate and compare the statistical behaviours of different measures when the dimension is high and the count data consist of many zeros.

## 3.1 EVALUATION INDEX

The main practical goal of DR is to eliminate noisy or uninformative dimensions of high-dimensional data and assist downstream classification/clustering algorithms in uncovering meaningful classes/clusters in the data. Different classes/clusters of count data can be characterized by different distributions, and thus a dissimilarity measure that distinguishes these distributions well could benefit the downstream analysis of the data when integrated into standard DR approaches. In this subsection, we propose an index to evaluate how well a dissimilarity measure separates those data points generated from different distributions and groups those from the same distribution. The definition of the proposed index is given in Definition 1.

**Definition 1** *Suppose there are two count data distributions, denoted by $F_{\mathbf{x}}$ and $F_{\mathbf{y}}$, respectively. Let $S_X = \{\mathbf{x}_1, \ldots, \mathbf{x}_{n_x}\}$ be the set of samples generated from $F_{\mathbf{x}}$ and $S_Y = \{\mathbf{y}_1, \ldots, \mathbf{y}_{n_y}\}$ the set of samples from $F_{\mathbf{y}}$. For a given dissimilarity measure $D(\cdot, \cdot)$, the proposed index $R(F_{\mathbf{x}}, F_{\mathbf{y}})$ is defined as*

$$\frac{\sum_{\mathbf{x} \in S_X, \mathbf{y} \in S_Y} D^2(\mathbf{x}, \mathbf{y})/(n_x n_y)}{\sum_{\mathbf{x}_i, \mathbf{x}_j \in S_x, \mathbf{x}_i \neq \mathbf{x}_j} \frac{D^2(\mathbf{x}_j, \mathbf{x}_i)}{(n_x - 1)n_x} + \sum_{\mathbf{y}_i, \mathbf{y}_j \in S_y, \mathbf{y}_i \neq \mathbf{y}_j} \frac{D^2(\mathbf{y}_j, \mathbf{y}_i)}{(n_y - 1)n_y}}. \quad (8)$$

Note that here we consider only two distributions for simplicity, and to facilitate the following analysis we use the squared dissimilarity function. In the following, the subscript $*$ of $R_*(F_{\mathbf{x}}, F_{\mathbf{y}})$ will be that of the corresponding dissimilarity measure. The mathematical objectives of many DR approaches are to preserve the global or local proximity of high-dimensional data, and the proposed index suits them in that $R(F_{\mathbf{x}}, F_{\mathbf{y}})$ assesses simultaneously the variation between data points from the same distribution (local proximity) and the separation between data points from different distributions (global proximity). $R(F_{\mathbf{x}}, F_{\mathbf{y}}) > 1$ implies that using the corresponding dissimilarity measure $D(\cdot, \cdot)$ makes the separation between $F_{\mathbf{x}}$ and $F_{\mathbf{y}}$ greater than the within-distribution variation. By construction, a higher value of $R(F_{\mathbf{x}}, F_{\mathbf{y}})$ would tend to indicate more powerful and robust discriminative ability of the corresponding measure in the presence of noisy dimensions, which possibly reduce the between-distribution separation and increase the within-distribution variation.

Before we dive into the comparison of measures using $R(F_{\mathbf{x}}, F_{\mathbf{y}})$, the statistical behaviour of $R(F_{\mathbf{x}}, F_{\mathbf{y}})$ in the high-dimensional space should be clarified. Proposition 1 presents the behaviour of $R(F_{\mathbf{x}}, F_{\mathbf{y}})$ for the dissimilarity functions in a generic form: $D^2(\mathbf{x}, \mathbf{y}) = \sum_{i=1}^p D^2(x_i, y_i) = \sum_{i=1}^p [f(x_i) - f(y_i)][g(x_i) - g(y_i)]$, which covers all the measures presented in Table 1. Proposition 1 shows that $R(F_{\mathbf{x}}, F_{\mathbf{y}})$ moves toward a constant as dimension $p$ grows, irrespective of the number of samples from distributions. The covariance of two increasing functions $(f, g)$ of a random variable is positive [Schmidt, 2003], and thus the constant which $R(F_{\mathbf{x}}, F_{\mathbf{y}})$ converges to would be greater than $\frac{1}{2}$ iff $[\mathbb{E}f(x) - \mathbb{E}f(y)][\mathbb{E}g(x) - \mathbb{E}g(y)] > 0$, which is readily satisfied in practice. The convergence still holds under some mild conditions, such as with dependent dimensions and non-identical distributions.

**Proposition 1** *Suppose points in $S_X \cup S_Y$ are independent, and each coordinate of $\mathbf{x}, \mathbf{y}$ in $S_X$ and $S_Y$ are independently drawn from $1$-dimensional non-degenerate data distributions $F_x$ and $F_y$, respectively. For $D^2(\mathbf{x}, \mathbf{y}) = \sum_{i=1}^p D^2(x_i, y_i) = \sum_{i=1}^p [f(x_i) - f(y_i)][g(x_i) - g(y_i)]$ with $x_i \sim F_x$, $y_i \sim F_y$, where $f(\cdot)$ and $g(\cdot)$ are predetermined functions, if $\mathbb{E}[D^2(x, y)]$, $\mathbb{E}[D^2(x, \tilde{x})]$ and $\mathbb{E}[D^2(y, \tilde{y})]$ exist for independent samples $\tilde{x}, x \sim F_x$, $\tilde{y}, y \sim F_y$, we have*

$$R_D(F_x, F_y) \overset{prob}{\to} \frac{1}{2} + \frac{1}{2} \frac{[\mathbb{E}f(x) - \mathbb{E}f(y)][\mathbb{E}g(x) - \mathbb{E}g(y)]}{\mathrm{Cov}(f(x), g(x)) + \mathrm{Cov}(f(y), g(y))},$$

*where $\overset{prob}{\to}$ denotes the convergence in probability as the dimension $p$ goes to infinity.*

## 3.2 COMPARE THE EUCLIDEAN DISTANCES W/O VSTS

In the following, we compare $D_E$ of original data with the Euclidean distances of the transformed data, according to

their behaviours when dealing with small counts in the high-dimensional space; that is, we compare them in terms of the respective constants that their $R\left(F_x, F_y\right)$'s converge to as the dimension diverges to infinity. We first examine the discriminative ability of $D_E$ when count data are small. Corollary 1 provides the sufficient and necessary condition for $R_E\left(F_x, F_y\right) \xrightarrow{p} c_E > 1$. This condition suggests that for any pairs of Poisson distributions that generate small counts with mean values less than 1, we obtain $c_E < 1$; that is, $D_E$ cannot distinguish the two distributions well. Therefore, $D_E$ is expected to perform poorly when handling small counts. An example showing $D_E$ is unable to distinguish two Poisson distributions with different patterns of small counts is provided in Section S.2 of Supplementary Material.

**Corollary 1** *With the same assumptions and notation as those in Proposition 1, for $D(\cdot, \cdot) = D_E(\cdot, \cdot)$, we have*

1. *$R_E\left(F_x, F_y\right) \xrightarrow{prob} c_E \geq \frac{1}{2}$ for some constant $c_E$. The equality holds iff $\mathbb{E}(x) = \mathbb{E}(y)$ for $x \sim F_x$, $y \sim F_y$.*
2. *$c_E > 1$ iff $[\mathbb{E}(x) - \mathbb{E}(y)]^2 > \text{Var}(x) + \text{Var}(y)$.*

We then investigate the behaviours of $R\left(F_x, F_y\right)$'s of the Euclidean distances based on VSTs when either $F_x$ or $F_y$ generates small counts. Without loss of generality, we assume $F_x$ produces small counts and $F_y$ is an arbitrary distribution. Suppose there is a VST characterized by an increasing transformation function $g(\cdot)$, such that $g(y) \geq 0$ for $y \geq 0$. Note that $g(\cdot)$ covers $g_r(y)$, $g_{\text{asin}}(y)$, and $g_{\log}(y)$. Let $D_E$ of the data transformed from raw counts by $g(\cdot)$ be $D_g$ and the corresponding index $R_g\left(F_x, F_y\right)$. Corollary 2 provides the difference between $c_g$ and $c_E$ when the proportion of zeros of each data point in $S_X$ moves toward 1. It shows that, as $\mu_x$ approaches 0, $D_g$ is better suited for distinguishing data points than $D_E$ iff $\frac{[g(0) - \mathbb{E}g(y)]^2}{\text{Var}[g(y)]} - \frac{\mathbb{E}^2(y)}{\text{Var}(y)} > 0$.

**Corollary 2** *Suppose that $x$ and $y$ are non-negative random variables. Let the expectation of $F_x$ be $\mu_x$. With the same assumptions and notation as those in Proposition 1, we have*

$$\lim_{\mu_x \to 0}\left(c_g - c_E\right) = \frac{1}{2}\left[\frac{[g(0) - \mathbb{E}g(y)]^2}{\text{Var}[g(y)]} - \frac{\mathbb{E}^2(y)}{\text{Var}(y)}\right],$$

*where $c_g$ and $c_E$ are the constants that $R_g\left(F_x, F_y\right)$ and $R_E\left(F_x, F_y\right)$ approach, respectively, as the dimension $p$ goes to infinity.*

To illustrate the advantages of applying VSTs to small counts, we obtain values of $\lim_{\mu_x \to 0} c_g = \frac{1}{2} + \frac{1}{2}\frac{[g(0) - \mathbb{E}g(y)]^2}{\text{Var}[g(y)]}$ for Poisson distributions $F_y$'s with different mean values and transformation functions by numerical computation. Figure 1 supplies the numerical results and shows that $g_r$,

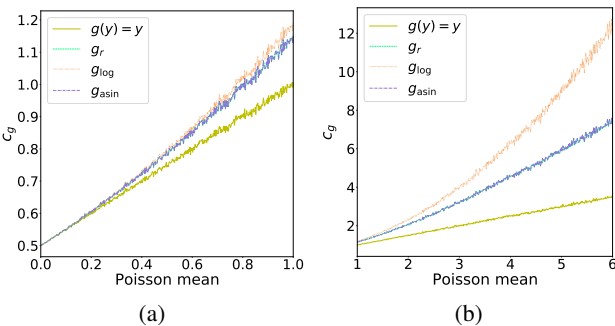

Figure 1: $c_g$ for different Poisson distributions and different transforms.

$g_{\log}$, and $g_{\text{asin}}$ always result in a $c_g$ that is no less than $c_E$. In particular, it is observed from Figure 1(a) that $c_g$'s exceed 1 when the Poisson mean is higher than 0.8, indicating that the corresponding measures distinguish better between data points with large proportions of zeros compared with $D_E$. Note that we assign a large value to $r$ in $c_{g_{\text{asin}}}$ ($r = 1000$) since an arbitrary $\text{NB}(r, p)$ approximates a Poisson distribution when $r$ approaches infinity. The above analysis suggests that, although the VSTs are unable to stabilize the variances of small count data, they improve the discriminative ability over $D_E$. Proofs of Proposition 1 and Corollary 2 are provided in Section S.1 of Supplementary Material.

### 3.3 COMPARE THE TWO PROPOSED MEASURES WITH OTHER DISSIMILARITY MEASURES

In the following, we will compare the proposed measures $(D_P, D_{NB})$ with the other dissimilarity measures in terms of the proposed index $R\left(F_x, F_y\right)$ computed in the high-dimensional space. Note that the estimate $\hat{R}(F_x, F_y)$ would be close enough to the constant that $R(F_x, F_y)$ approaches as long as the dimension is high enough. Take a pair of distributions $(F_x, F_y)$ and a pair of measures $(D_{NB}, D_E)$ for example, we believe $D_{NB}$ is superior to $D_E$ for distinguishing between $F_x$ and $F_y$ if $\hat{R}_{NB}(F_x, F_y) > \hat{R}_E(F_x, F_y)$. Further, to thoroughly evaluate their discriminative abilities for a specific distribution type, we compare their performances in terms of the fraction that $\hat{R}_{NB}(F_x, F_y) > \hat{R}_E(F_x, F_y)$ for different configurations of parameters. The fraction greater than 0.5 suggests $D_{NB}$ is better suited for distinguishing this distribution type than $D_E$ and vice versa. The distributions types $(F_x, F_y)$ taken into account are the broadly used Poisson and NB distributions. $F_x$ and $F_y$ are of the same distribution type but with different parameter configurations. It is worth mentioning that $R_P(F_x, F_y)$ and $R_{NB}(F_x, F_y)$ are the same when data are Poisson-distributed, because $D_{NB}$ approaches $D_P$ when the dispersion parameter $r$ goes to infinity. We thus exclude $D_{NB}$ from the comparison for Poisson-distributed data. More

Table 2: Fraction that $\hat{R}(F_x, F_y)$ of a measure is greater than that of another measure for Poisson distributions. The value in entry $(i, j)$ represents the fraction that $\hat{R}$ of the measure on the $i$-th row is greater than that of the measure on the $j$-th column. Top two measures are shown in bold.

| Measures | $D_E$ | $D_r$ | $D_{\text{asin}}$ | $D_{\log}$ | $D_P$ | Ave |
|---|---|---|---|---|---|---|
| $D_E$ | - | 0.060 | 0.060 | 0.060 | 0.040 | 0.055 |
| $D_r$ | 0.940 | - | 0.010 | 0.080 | 0.080 | 0.295 |
| $D_{\text{asin}}$ | 0.940 | 0.940 | - | 0.010 | 0.010 | 0.520 |
| $D_{\log}$ | 0.940 | 0.900 | 0.900 | - | 0.520 | **0.815** |
| $D_P$ | 0.960 | 0.920 | 0.900 | 0.480 | - | **0.815** |

Table 3: Fraction that $\hat{R}(F_x, F_y)$ of a measure is greater than that of another measure for NB distributions.

| Measures | $D_E$ | $D_r$ | $D_{\text{asin}}$ | $D_{\log}$ | $D_P$ | $D_{NB}$ | Ave |
|---|---|---|---|---|---|---|---|
| $D_E$ | - | 0.240 | 0.280 | 0.242 | 0.056 | 0.050 | 0.174 |
| $D_r$ | 0.760 | - | 0.534 | 0.498 | 0.058 | 0.054 | 0.381 |
| $D_{\text{asin}}$ | 0.720 | 0.466 | - | 0.352 | 0.106 | 0.052 | 0.339 |
| $D_{\log}$ | 0.758 | 0.502 | 0.648 | - | 0.112 | 0.056 | 0.415 |
| $D_P$ | 0.944 | 0.942 | 0.894 | 0.888 | - | 0.070 | **0.748** |
| $D_{NB}$ | 0.950 | 0.946 | 0.948 | 0.944 | 0.930 | - | **0.944** |

details on the simulations and the calculation of different measures are presented in Section S.3 of Supplementary Material.

After simulations, we get $\hat{R}(F_x, F_y)$'s for a wide scope of parameter configurations and different distribution types. As mentioned before, the measures are compared in terms of $\hat{R}(F_x, F_y)$'s. Table 2 and Table 3 supply the comparisons of the measures when data follow Poisson and NB distributions, respectively. For small count data following Poisson distributions, $D_P/D_{NB}$ performs as well as $D_{\log}$. Furthermore, $D_{NB}$ is superior to the other measures when count data are negative-binomially distributed. $D_E$, as anticipated, performs much worse than the other measures. The simulation results show that $D_{NB}$ is better than the other measures when distinguishing Poisson and NB distributions, and thus we expect that $D_{NB}$ outperforms the others when integrated into standard DR methods.

Although it is shown that $D_{NB}$ is superior to the other measures and $D_P$ is the second-best measure, we find that the calculation of $m_i$ affects their discriminative performance. Specifically, if the value of $m_i$ is closer to the average of the expected values of the two distributions $(F_x, F_y)$, $D_P$ and $D_{NB}$ would have a better discriminative ability. We reason that the mean of the expected values can be regarded as a typical value informative to the parameter estimation when used in the priors and thus is beneficial for the calculation of the KL divergence.

# 4 EXPERIMENTAL RESULTS

In this section, we present experimental results of representative DR methods with different dissimilarity mea-

Table 4: Real scRNA-seq datasets used in this paper.

| Dataset | #clusters | #cells | #genes | prop of zeros |
|---|---|---|---|---|
| sc-CEL-seq2 [Tian et al., 2019] | 3 | 274 | 22060 | 0.678 |
| sc-CEL-seq2-5cl-p1 [Tian et al., 2019] | 5 | 297 | 15564 | 0.608 |
| sc-CEL-seq2-5cl-p2 [Tian et al., 2019] | 5 | 307 | 14078 | 0.598 |
| sc-CEL-seq2-5cl-p3 [Tian et al., 2019] | 5 | 305 | 13426 | 0.643 |
| Zheng8eq [Zheng et al., 2017] | 8 | 3994 | 13301 | 0.957 |

Table 5: Simulated scRNA-seq datasets used in this paper.

| Dataset | #clusters | #cells | #genes | prop of zeros | corresponding real dataset |
|---|---|---|---|---|---|
| sim-Zheng8eq | 8 | 3994 | 13770 | 0.969 | Zheng8eq [Zheng et al., 2017] |
| sim-manno-vm | 5 | 1977 | 19416 | 0.899 | manno-ESCs [La Manno et al., 2016] |
| sim-manno-ESCs | 5 | 1715 | 19459 | 0.834 | manno-ventral-midbrain [La Manno et al., 2016] |

sures on both real and simulated high-dimensional count datasets with large fractions of zeros. In addition, we compare the generalised PCAs (GPCAs) [Collins et al., 2002] and NMF [Dhillon and Sra, 2005] with the proposed measure $D_{NB}$ in Section S.6 of Supplementary Material.

## 4.1 DATASETS

The high-dimensional count data considered in this paper is the single cell RNA sequencing (scRNA-seq) data with unique molecular identifiers (UMI). scRNA-seq data offer a unique opportunity to investigate the stochastic heterogeneity of complex issues at a near-genome-wide scale [Saliba et al., 2014, Shapiro et al., 2013, Kolodziejczyk et al., 2015]. scRNA-seq data with UMI are often modelled by NB or Poisson distributions [Townes et al., 2019, Kim et al., 2020, Svensson, 2020] and exhibit large proportions of zero counts. We run experiments on both real and simulated scRNA-seq datasets. These datasets contain large proportions of zeros, ranging from $0.6$ to $0.97$. The characteristics of the real scRNA-seq datasets used in this paper are summarized in Table 4. Cluster labels provided by the real scRNA-seq datasets correspond to different cell types: labels for the datasets obtained from Tian et al. [2019] are assigned in terms of cancer cell lines and those for the Zheng8eq dataset based on the types of purified peripheral blood mononuclear cells. All the cluster labels reported in these datasets are defined independently of gene expression profiles and can be used as the ground-truth labels. We simulate three additional scRNA-seq datasets by using the R's Splatter package [Zappia et al., 2017] with most of the parameters learned from real datasets except for differential expression factors, which determine the difference between groups of cells and the number of clusters. The information of the simulated datasets and the corresponding real datasets used for simulations are summarized in Table 5.

## 4.2 EVALUATION

**Representative DR methods.** We compare different measures with three representative DR methods: PCA, GPLVM, and tSNE. GPLVM and PCA seek to retain the global struc-

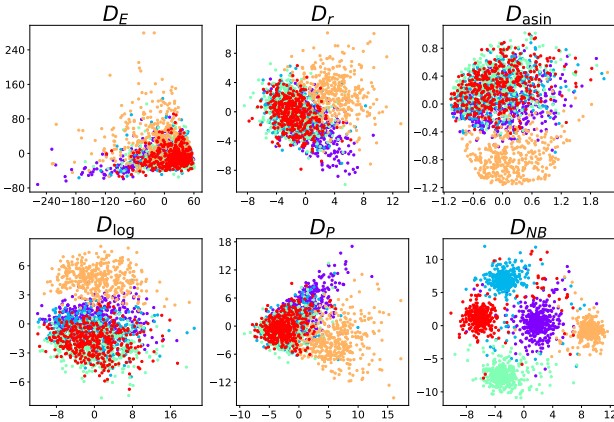

Figure 2: Visualization of the sim-manno-vm dataset obtained by GPLVMs with different measures. Different clusters are shown in different colours.

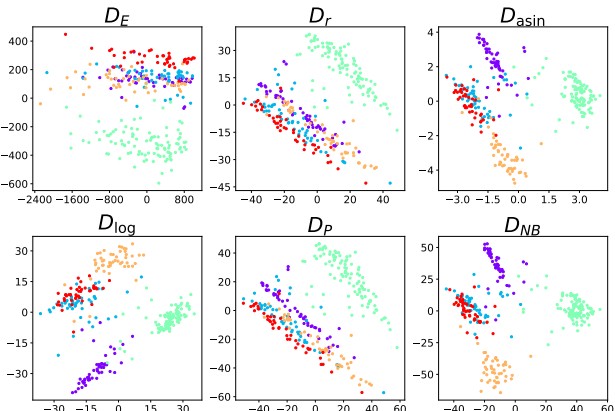

Figure 3: Visualization of the sc-CEL-seq2-5cl-p1 dataset obtained by PCA with different measures.

ture of data by preserving the pairwise proximity for all pairs of data points, while tSNE predominantly preserves the local structure with the pairwise proximity amongst neighbouring data points. Therefore, DR results presented by PCA/GPLVM and tSNE, respectively, are complementary. The proposed measures and $D_E$'s of the data transformed by the VSTs are compared based on their performance when integrated into the DR approaches. Note that $r$ in $D_{NB}$ and $D_{\mathrm{asin}}$ is set to the common NB dispersion parameter estimated by the R's edgeR package [Robinson et al., 2009]. As we treat mean values of features as pseudo observations when deriving the proposed measures, we replace each value $x$ in the data matrix with $\frac{x+m}{2}$, where $m$ is the mean value of the corresponding feature column, when estimating $r$.

**Visualization.** As visualization is an important application of DR, we evaluate the DR methods with different dissimilarity measures by visually inspecting their DR results in a two-dimensional (2D) space. A good visualization should exhibit well-separated groups of data.

**Clustering.** Apart from visualization in a 2D space, DR techniques can also be used for improving clustering of high-dimensional data in real-world applications. For instance, high-dimensional scRNA-seq data are often projected into a low-dimensional space whose dimension could be greater than 2, and clustering methods, such as $k$-means and hierarchical clustering, are performed on the dimension-reduced data to improve clustering [Sun et al., 2019, Petegrosso et al., 2020]. Furthermore, it has been shown that applying $k$-means clustering in a PCA subspace can significantly improve clustering accuracy [Ding and He, 2004]. As clustering is an important downstream task to DR, we also evaluate the DR approaches with different measures based on the clustering performance in the dimension-reduced space.

The $k$-means algorithm [MacQueen, 1967, Lloyd, 1982] is used for inferring the cluster labels of data in the space of reduced dimension. The number of clusters in the $k$-means algorithm is set to be the ground truth. The clustering performance is assessed in terms of the adjusted rand index (ARI) [Hubert and Arabie, 1985] between the cluster labels from the original publication/simulation and the inferred ones. The higher the ARI, the better the performance. Normally, tSNE and GPLVM map high-dimensional data to a 2D space, and thus we consider only 2D projections when evaluating their clustering performance. Since the results of the tSNE algorithm could be variable, we replicate the procedure, of first performing tSNE and then applying $k$-means, for 10 times on the datasets for a more reliable comparison. More experimental details are provided in Section S.4 of Supplementary Material.

### 4.3 VISUALIZATION

In this subsection, we examine whether the application of $D_{NB}$ can produce better visualization.

First, we visualize the dimension-reduced data obtained by GPLVMs with different measures in Figure 2 and Figures S2-S8 of Supplementary Material. It is observed that GPLVMs with $D_{NB}$, $D_r$, $D_{\mathrm{asin}}$, and $D_{\log}$ perform equally well on most datasets except for the sim-manno-ESCs and sim-manno-vm datasets. For the sim-manno-ESCs dataset (Figure S8 of Supplementary Material), GPLVMs with $D_P$ and $D_{NB}$ display well-grouped data in the 2D space while those with the other measures fail to do so. Furthermore, only the GPLVM with $D_{NB}$ can distinguish different groups of data points on the sim-manno-vm dataset (Figure 2).

Secondly, we compare the visualization results obtained by tSNE with different measures in Figures S9-S16 of Supplementary Material. The tSNE algorithms with $D_{NB}$, $D_r$, $D_{\mathrm{asin}}$, and $D_{\log}$ produce well-distinguished groups of data on most datasets except for the Zheng8eq dataset (Figure S13), the sim-manno-ESCs dataset (Figure S15) and the sim-manno-vm dataset (Figure S16), where all the measures

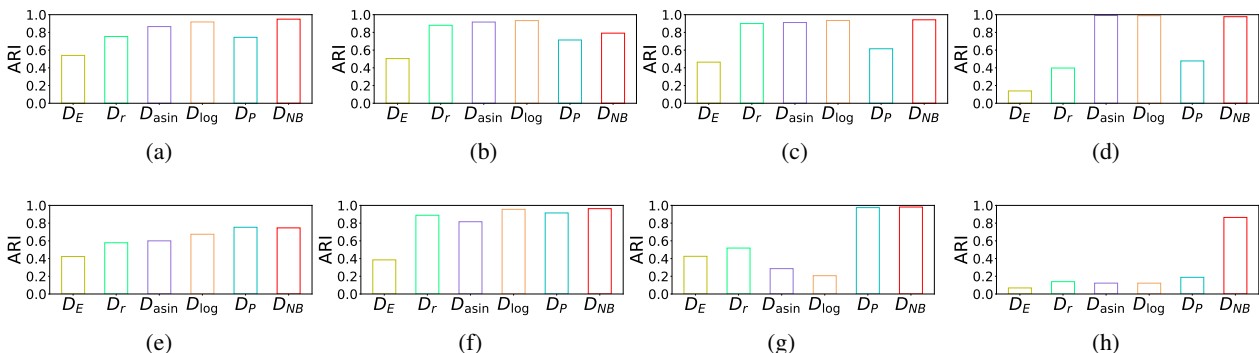

Figure 4: ARI of $k$-means with GPLVMs and different dissimilarity measures on the following datasets: (a) sc-CEL-seq2-5cl-p1, (b) sc-CEL-seq2-5cl-p2, (c) sc-CEL-seq2-5cl-p3, (d) sc-CEL-seq2, (e) Zheng8eq, (f) sim-Zheng8eq, (g) sim-manno-ESCs, and (h) sim-manno-vm.

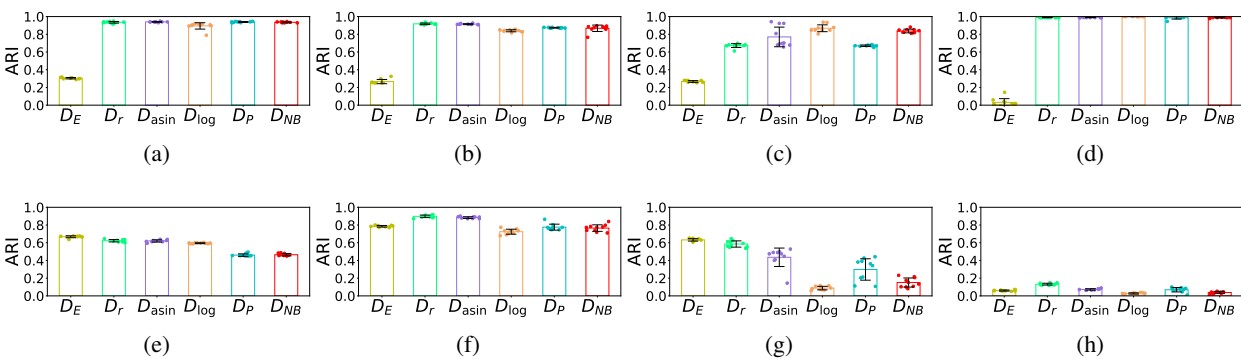

Figure 5: ARI of $k$-means with tSNE and different dissimilarity measures on the following datasets: (a) sc-CEL-seq2-5cl-p1, (b) sc-CEL-seq2-5cl-p2, (c) sc-CEL-seq2-5cl-p3, (d) sc-CEL-seq2, (e) Zheng8eq, (f) sim-Zheng8eq, (g) sim-manno-ESCs, and (h) sim-manno-vm.

fail to recognize the groups of data.

Thirdly, by comparing PCA results with different measures in the 2D space (Figure 3 and Figures S17-S23 of Supplementary Material), we find that the PCA algorithms with $D_{NB}$, $D_{asin}$, $D_{log}$ produce more distinguished groups of data on the sc-CEL-seq2-5cl-p1 dataset (Figure 3) and the sc-CEL-seq2-5cl-p2 dataset (Figure S17). For the sc-CEL-seq2-5cl-p3 dataset (Figure S18), the PCA with $D_{log}$ presents three distinguished groups while those with the other measures display only two groups. The PCA with $D_{NB}$ and $D_{asin}$ can separate the groups in the sc-CEL-seq2 dataset (Figure S19). The PCA algorithms with the proposed measures and VSTs perform equally well on the Zheng8eq dataset (Figure S20) and the sim-Zheng8eq dataset (Figure S21). For the sim-manno-ESCs dataset (Figure S22), the applications of $D_r$, $D_P$, and $D_{NB}$ lead to more distinguished groups in the data. Furthermore, it is observed in Figure S23 that only the PCA with $D_{NB}$ can separate groups of data to some degree on the sim-manno-vm dataset while those with the other measures fail to do so.

It is found that the GPLVM/PCA with $D_{NB}$ often presents distinguished groups of data in the 2D space while the tSNE with $D_{NB}$ fails to do so on some datasets. This difference may be due to the characteristics of the DR methods, but not the dissimilarity measures themselves. As mentioned before, GPLVM/PCA aims to preserve the global structure of data and tSNE predominantly preserves the local structure. Thus, tSNE may fail to preserve the global structure (inter-cluster proximity) due to its preference for the local proximity. In such cases, visualizing clusters with tSNE would not work well. Although Kobak and Berens [2019] suggest using informative initialization or multi-scale similarities to improve the preservation of the global structure, but we find that these strategies do not result in better-distinguished clusters in the 2D space for the Zheng8eq, sim-manno-ESCs and sim-manno-vm datasets.

To sum up, the GPLVM and PCA with $D_{NB}$ can produce better visualization results than those with the other measures, while the tSNE with $D_{NB}$ may not distinguish clusters well due to its preference for the local structure of data.

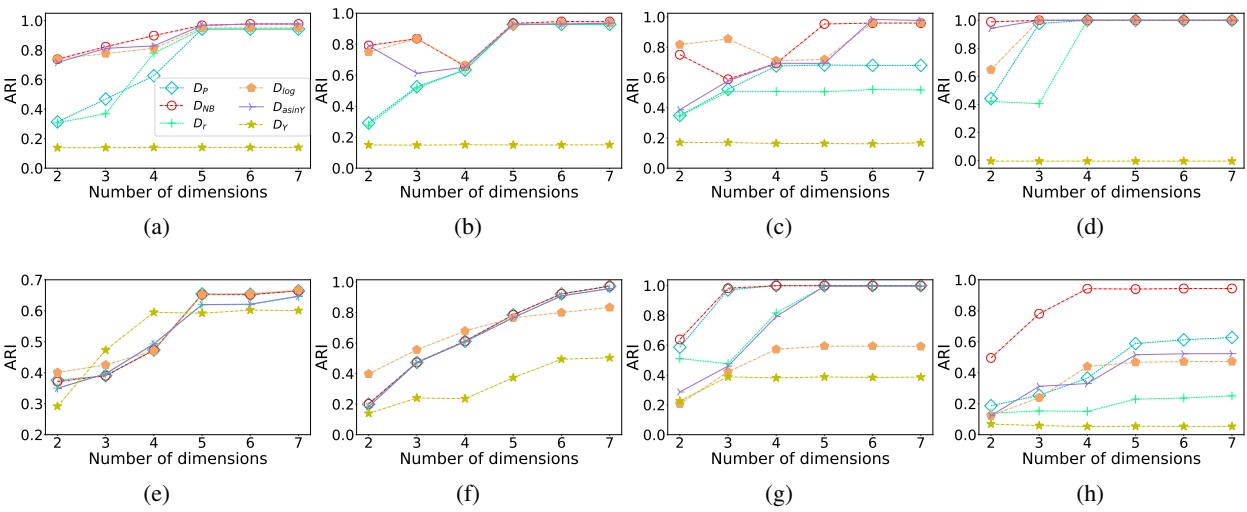

Figure 6: ARI of $k$-means with PCA and different dissimilarity measures on the following datasets: (a) sc-CEL-seq2-5cl-p1, (b) sc-CEL-seq2-5cl-p2, (c) sc-CEL-seq2-5cl-p3, (d) sc-CEL-seq2, (e) Zheng8eq, (f) sim-Zheng8eq, (g) sim-manno-ESCs, and (h) sim-manno-vm.

## 4.4 CLUSTERING RESULTS

In the following, we compare different measures in terms of their clustering performance in the dimension-reduced spaces. The $k$-means clustering results with GPLVMs and different dissimilarity measures are shown in Figure 4. It is observed that the GPLVM with $D_{NB}$ performs consistently well on most datasets except the sc-CEL-seq2-5cl-p2. Furthermore, the GPLVM with $D_{NB}$ obtains much higher values than the other measures based on VSTs on the sim-manno-ESCs dataset and the sim-manno-vm dataset. To sum up, $D_{NB}$ outperforms the other measures when integrated into GPLVM.

Figure 5 presents the $k$-means clustering results with tSNE and different measures. The tSNE with $D_{NB}$ performs comparably well on some datasets except for the Zheng8eq, sim-manno-ESCs, and sim-manno-vm datasets. The clustering performance of the tSNE with $D_{NB}$ are consistent with its visualization results. As we discussed before, its relatively weak performance on the Zheng8eq, sim-manno-ESCs, and sim-manno-vm datasets may be due to the tSNE's preference for the preservation of the local structure of data. Furthermore, the clustering performance of tSNE suggests that clustering on the outputs of DR techniques must be done with caution. DR approaches, such as non-linear tSNE, could be unsuccessful to preserve clusters and thus adversely affect the cluster analysis.

It is observed in Figure 6, $D_{NB}$ is superior to the other measures when combined with PCA, irrespective of the number of dimensions, on most datasets except for the sc-CEL-seq2-5cl-p3, Zheng8eq, and sim-Zheng8eq datasets. On these three datasets, $D_{NB}$ outperforms most measures

when the dimension is greater than five.

In summary, consistent with the visualization results, the clustering performance obtained by the representative GPLVM/PCA with $D_{NB}$ are superior to those with the other measures.

## 5 CONCLUSION

This paper investigates how to perform DR for high-dimensional small count data. We propose a dissimilarity measure $D_{NB}$ that is well-suited for count data with many zeros. The statistical behaviours of different dissimilarity measures when the dimension is high enough are investigated in terms of a proposed index. It is found that the proposed measure $D_{NB}$ is superior to the other measures in the sense that it distinguishes data points from different distributions better. Consistent with the statistical comparison, the experimental results demonstrate that $D_{NB}$ enhances a variety of standard DR approaches.

### Acknowledgements

The authors thank the anonymous reviewers for helpful discussions and suggestions.

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
