# OpenReview forum: "Dimension Reduction for High-dimensional Small Counts with KL Divergence"
_auai.org/UAI/2022/Conference — UAI 2022 Poster_

### Official Review · Reviewer_mEXs · 2022-03-27

**Q2(1) Originality/Novelty:** 3
**Q2(2) Significance/Impact:** 3
**Q2(3) Correctness/Technical Quality:** 3
**Q2(6) Clarity Of Writing:** 3
**Q6 Overall Score:** 7
**Q8 Confidence In Your Score:** 3

**Q1 Summary And Contributions:**

This paper takes both Poisson and negative binomial (NB) distributions into account and proposes two dissimilarity measures for small count data based on the Kullback-Leibler (KL) divergence. This paper also proposes an index to evaluate the discrimination abilities of different dissimilarity measures and shows their proposed dissimilarity measures can achieve superior discrimination ability compared with other widely used dissimilarity.

**Q2 Assessment Of The Paper:**

More detailed information regarding each of these aspects is given below:

**Q2(4) Quality Of Experiments (Optional):**

3: Good: The experimental evaluation is adequate, and the results convincingly support the main claims.

**Q2(5) Reproducibility:**

3: Good: Key resources (e.g., proofs, code, data) are available and key details (e.g., proofs, experimental setup) are sufficiently well-described for competent researchers to confidently reproduce the main results.

**Q3 Main Strengths:**

In this paper, the authors mainly focus on the dimension reduction techniques on high-dimensional small count data. Specifically, the authors took both Poisson and negative binomial (NB) distributions into account and proposed two dissimilarity measures for small count data based on the Kullback-Leibler (KL) divergence. In addition, they used empirical Bayes estimators to estimate the distributional parameters so as to calculate the KL divergence reliably. They also proposed an index to evaluate the discrimination abilities of different dissimilarity measures and show their proposed dissimilarity measures can achieve superior discrimination ability compared with other widely used dissimilarity.

**Q4 Main Weakness:**

1) From the results in Tables 2-3 and the description on the right column of Page 5, for small count data following Poisson distributions, D_{log} achieves better performances than D_{NB}. Please give more detailed explanations and analysis why the proposed D_{NB} can not achieve the same good results than D_{log} in these scenarios.
2) The length of appendixes in this paper is too long. More important, many results and analysis in Section 4 rely on the figures in appendixes. I suggest authors show the important results in the main body of the paper, and add more detailed analysis about the results shown in the main body rather than appendixes.
3) There are some evident typos in the paper. For example, on the left column of Page 3, three lines above Equation 4, Pois(\lamda_{ix}) should change to Pois(\lamda_{iy}). On the second line on the right column of Page 4, F_{x} and F_{x} should change to F_{x} and F_{y}.


**Q5 Detailed Comments To The Authors:**

In this paper, the authors mainly focus on the dimension reduction techniques on high-dimensional small count data. Specifically, the authors took both Poisson and negative binomial (NB) distributions into account and proposed two dissimilarity measures for small count data based on the Kullback-Leibler (KL) divergence. In addition, they used empirical Bayes estimators to estimate the distributional parameters so as to calculate the KL divergence reliably. They also proposed an index to evaluate the discrimination abilities of different dissimilarity measures and show their proposed dissimilarity measures can achieve superior discrimination ability compared with other widely used dissimilarity.

In my opinion, the motivation of this paper is clear, and the proposed dissimilarity measures seem like reasonable, but the writing of this paper still needs to improve. Some suggestions are listed as follows:
1) From the results in Tables 2-3 and the description on the right column of Page 5, for small count data following Poisson distributions, D_{log} achieves better performances than D_{NB}. Please give more detailed explanations and analysis why the proposed D_{NB} can not achieve the same good results than D_{log} in these scenarios.
2) The length of appendixes in this paper is too long. More important, many results and analysis in Section 4 rely on the figures in appendixes. I suggest authors show the important results in the main body of the paper, and add more detailed analysis about the results shown in the main body rather than appendixes.
3) There are some evident typos in the paper. For example, on the left column of Page 3, three lines above Equation 4, Pois(\lamda_{ix}) should change to Pois(\lamda_{iy}). On the second line on the right column of Page 4, F_{x} and F_{x} should change to F_{x} and F_{y}.


**Q7 Justification For Your Score:**

In my opinion, the motivation of this paper is clear, and the proposed dissimilarity measures seem like reasonable, but the writing of this paper still needs to improve.

**Q9 Complying With Reviewing Instructions:**

1: Yes.

---

### Official Review · Reviewer_yrEp · 2022-04-01

**Q2(1) Originality/Novelty:** 3
**Q2(2) Significance/Impact:** 2
**Q2(3) Correctness/Technical Quality:** 3
**Q2(6) Clarity Of Writing:** 3
**Q6 Overall Score:** 6
**Q8 Confidence In Your Score:** 2

**Q1 Summary And Contributions:**

The authors propose two new measures for dimension reduction (DR) on count data with high sparsity and small entries. They also introduce a metric for comparing different DR approaches, and evaluate several existing measures along with their novel measures, in conjunction with 3 standard DR methods.

**Q2 Assessment Of The Paper:**

More detailed information regarding each of these aspects is given below:

**Q2(4) Quality Of Experiments (Optional):**

4: Excellent: The experimental evaluation is comprehensive and the results are compelling.

**Q2(5) Reproducibility:**

3: Good: Key resources (e.g., proofs, code, data) are available and key details (e.g., proofs, experimental setup) are sufficiently well-described for competent researchers to confidently reproduce the main results.

**Q3 Main Strengths:**

The authors provide a strong conceptual foundation for the measures they propose, and validate their expectations with both simulated and real data. Numerical assessments confirm the conclusions suggested by visual inspection.

**Q4 Main Weakness:**

My only concern is that the paper does not devote sufficient attention to the results achieved, Since the newly-proposed D_{NB} performs so well on certain data sets and only reasonably well on others, it would probably be quite informative to investigate the specific characteristics of the contrasting conditions that are responsible for these differences. (In fact, such an investigation could possibly also give useful intuitions on the applicability of, say GPLVM vs tSNE.)


**Q5 Detailed Comments To The Authors:**

The paper is well thought out and easy to read, even for someone like myself who is not an expert in this field. I caught a few minor issues in the wording, and list those below.
- Last paragraph of Sec 1: 'discriminate ability' should be 'discriminative ability'.
- The statement 'It is straightforward to see that the higher the value of R(Fx,Fy), the more powerful the discrimination ability of the dissimilarity measure.' seems too strong: for example, a non-linear transformation of a given D may enhance R without affecting discrimination. I would suggest something like 'By construction, a higher value of R(Fx,Fy) will tend to indicate more powerful discrimination ability'
- The relatively weak performance of D_P for data that is truly Poisson distributed (Table 2) needs some explanation.
- In Sec 4,3, I suggest replacing 'the uses of Dr, DP , and DNB lead to more ..' with something like 'applications of Dr, DP , and DNB lead to more ..'

**Q7 Justification For Your Score:**

I found the paper to be credible and informative - my only reservation is that its impact may not be substantial. Then again, I do not have detailed expertise in this matter, and my opinion should therefore not be given too much weight.


**Q9 Complying With Reviewing Instructions:**

1: Yes.

---

### Official Review · Reviewer_5i27 · 2022-04-10

**Q2(1) Originality/Novelty:** 2
**Q2(2) Significance/Impact:** 2
**Q2(3) Correctness/Technical Quality:** 3
**Q2(6) Clarity Of Writing:** 3
**Q6 Overall Score:** 4
**Q8 Confidence In Your Score:** 3

**Q1 Summary And Contributions:**

This paper proposes to use negative binomial (NB) distribution and the KL divergence to model count data and to perform dimensionality reduction. The author first used the KL divergence between the posterior (with respect to some prior) to tackle the numerical stability of sparse count data. Then the authors used a Fisher discriminant analysis type criterion and showed that the proposed distance $D_{NB}$ is the best among a set of distance measures.

**Q2 Assessment Of The Paper:**

More detailed information regarding each of these aspects is given below:

**Q2(4) Quality Of Experiments (Optional):**

3: Good: The experimental evaluation is adequate, and the results convincingly support the main claims.

**Q2(5) Reproducibility:**

3: Good: Key resources (e.g., proofs, code, data) are available and key details (e.g., proofs, experimental setup) are sufficiently well-described for competent researchers to confidently reproduce the main results.

**Q3 Main Strengths:**

This paper tackles an important problem to define an information-theoretical distance of high dimensional count data for dimensionality reduction. The problem setting suits well in the UAI conference.

The quality of writing is satisfactory.

**Q4 Main Weakness:**

Overall, the novelty is limited. The information theoretical measurements have been used for dimensionality reduction. See for example (Carter et al. 2011). The proposed measurements are mainly instances of methods that are already known. The distance measures listed in table 1 are simply KL divergences between Poisson/NB posterior distributions. It is expected that the Poisson/NB distribution can better model count data, especially in the posterior setting.

Kevin M. Carter; Raviv Raich; William G. Finn; Alfred O. Hero,III. Information-Geometric Dimensionality Reduction. 2011.

The evaluation metric $R(F_x,F_y)$ corresponds to the discrimination power of the distance measure with respect to two given distributions. It is not clear how it aligns with the objective of dimensionality reduction. In dimensionality reduction, we usually consider the spectrum of the pairwise proximity matrix. I am not fully convinced that $R(F_x, F_y)$ is a good measure for dimensionality reduction.

**Q5 Detailed Comments To The Authors:**

introduce the abbreviation VST at its first appearance

eq.(1) extra "( )"

Proposition 1 explain the notation "$\stackrel{p}{\to}$"

Corollary 2, suddenly here, the statement uses the $\lim$ notations. Please unify the $\to$ notations.

section 3.3
"Given a pair of distributions (Fx , Fy) and that of measures" is grammarly incorrect.

section 4.3 heavily refers to the appendix. This is hard to read. Please move these contents into the appendix or move the figures into the main text.

** After Rebuttal **

I have read the authors' responses and had another look at the paper. Embedding (or performing dimensionality reduction on) probability distributions is not new. The significance of the proposed dissimilarity measure in section 2 lies on applying existing techniques to new data (small counts).

Despite the good performance reported, the proposed distance measure has more parameters that correspond to the priors. See table 1 in the paper.

This distance is assessed based on the authors' definition in section 3.1. I can agree that it is a reasonable definition. However, to be more complete, the evaluation of different distances should be based on some other evaluation method (such as the spectrum of the pairwise distances), besides the authors' own evaluation method.

Overall, I won't be disappointed if the paper is accepted but recommend the authors to have a revision in either case.

**Q7 Justification For Your Score:**

The method uses KL divergence between two posteriors to measure the distance between two count data. It is useful in practice but does not advance the state-of-the-art of such count data modeling. The significance may not meet the UAI standard.

**Q9 Complying With Reviewing Instructions:**

1: Yes.

---

### Official Review · Reviewer_8A5W · 2022-04-13

**Q2(1) Originality/Novelty:** 3
**Q2(2) Significance/Impact:** 3
**Q2(3) Correctness/Technical Quality:** 3
**Q2(6) Clarity Of Writing:** 3
**Q6 Overall Score:** 7
**Q8 Confidence In Your Score:** 4

**Q1 Summary And Contributions:**

The paper presents new dissimilarity measures for count data.  Standard dissimilarity measures suffer from poor estimators when count data include zeros; the proposed measures remedy to this problem.
The paper also compares the measures in their ability to discriminate between data from two different distributions.


**Q2 Assessment Of The Paper:**

More detailed information regarding each of these aspects is given below:

**Q2(4) Quality Of Experiments (Optional):**

3: Good: The experimental evaluation is adequate, and the results convincingly support the main claims.

**Q2(5) Reproducibility:**

3: Good: Key resources (e.g., proofs, code, data) are available and key details (e.g., proofs, experimental setup) are sufficiently well-described for competent researchers to confidently reproduce the main results.

**Q3 Main Strengths:**

Standard distances or dissimilarity measures are too often used regardless of the specific nature of the data.  This paper deals with small count data, and opens the way to use novel dissimilarity measures more appropriate to such data in a wide range of tools and applications.

**Q4 Main Weakness:**

The second paper of the paper intrinsically requires the 9 pages of additional material, which is mostly gathered in appendixes for paper length reasons.  Furthermore, even is the dimensionality reduction part of the paper is an important application of the proposed dissimilarity measures, this part of the paper is less convincing: it mixes linear and nonlinear methods (whose goals are quite different), and it seems to make the assumption that DR methods can be evaluated by clustering performances, which is rarely the case on real-world data.

**Q5 Detailed Comments To The Authors:**

The paper introduces new dissimilarity measures for small count data.  This part of the paper is innovative (to my knowledge), very well written and may open the path to a wide range of algorithms and applications of these dissimilarity measures.  Although dimensionality reduction is an excellent application of these measures and should certainly not be removed from the paper, the link between the dissimilarity measures, clustering, and dimensionality reduction, is not clear enough.  Clustering could be performed without dimensionality reduction, while the latter is mainly intended for visualization.  In addition, the paper uses both linear and non-linear DR methods, which is confusing: on most cases linear DR methods will decrease the quality of clustering, while in nonlinear DR methods this decrease is largely influenced by the tricky metaparameters that have to be adjusted (for example in t-SNE); there exist now better methods that integrate all scales in a single DR and that are therefore much less sensitive to metaparameters.  Given that it seems obvious that the authors had to rely on the supplementary material to increase the length of the paper beyond 10 pages (making the 10-pages paper itself hardly self-contained), I suggest to concentrate the paper on the dissimilarity measures, and shorten the part on clustering and DR which may be seen as two (possibly independent) applications of the new measures.

**Q7 Justification For Your Score:**

Excellent paper, innovative and useful proposals.

**Q9 Complying With Reviewing Instructions:**

1: Yes.

---

### Decision · Program_Chairs · 2022-05-15

**Decision:**

Accept (Poster)

**Comment:**

Meta Review: Accounting for the presence of many zeros in dimensionality reduction for count data is an important topic.  The AC and reviewers all agree that this paper makes some pertinent contributions to that problem. For such contribution to be fully clear and impactful, we strongly urge the authors to follow up on their promise to incorporate several important points in their revision, including:
- clarifying the link between dissimilarity measures, clustering and DR by incorporating the points the authors made in their feedback to reviewer  8A5W, and as the reviewer suggested focusing on the dissimilarity measure portion while shortening the 2 applications to make the paper more self contained.
- clarifying the motivation of using the proposed R index to evaluate dissimilarity measures as opposed to the  spectrum of the pairwise distances)
- incorporating the new results provided as feedback to Reviewer yrEp